# Mixed Metal Phosphonates: Structure and Proton Conduction Manipulation through Various Alkaline Earth Metal Ions

Song-Song Bao *, Nan-Zhu Li, Yu-Xuan Wu and Yang Shen

State Key Laboratory of Coordination Chemistry, School of Chemistry and Chemical Engineering, Nanjing University, Nanjing 210023, China
*   Correspondence: baososo@nju.edu.cn

**Abstract:** Three new layered mixed metal phosphonates $[CoMg(notpH_2)(H_2O)_2]ClO_4 \cdot nH_2O$ (**CoMg·nH_2O**), $[Co_2Sr_2(notpH_2)_2(H_2O)_5](ClO_4)_2 \cdot nH_2O$ (**CoSr·nH_2O**), and $[CoBa(notpH_2)(H_2O)_{1.5}]ClO_4$ (**CoBa**) were synthesized by reacting a tripodal metalloligand $Co^{III}(notpH_3)$ $[notpH_6 = C_9H_{18}N_3(PO_3H_2)_3]$ with alkaline earth metal ions. Along with an increase in the radius of the alkaline earth metal ions, the 6-coordinate $\{MgO_6\}$, 7-coordinate $\{SrO_7\}$, and 9-coordinate $\{BaO_9\}$ geometries are the distorted octahedron, capped triangular prism, and tricapped triangular prism, respectively. Consequently, the metalloligand $Co(notpH_2)^-$ adopts variable coordination modes to bind the alkaline earth metal nodes, forming diverse layer topologies in the three mixed metal phosphonates. The AC impedance measurements revealed that the proton conductivities at 25 °C and 95% relative humidity (RH) follow the sequence: **CoMg·nH_2O > CoSr·nH_2O > CoBa**. As expected, **CoMg·nH_2O** exhibits a 28-fold enhanced value for proton conductivity ($4.36 \times 10^{-4}$ S cm$^{-1}$) compared with the previously reported isostructural compound, **CoCa·nH_2O**, at 25 °C and 95% RH due to the greater Lewis acid strength of Mg(II) lowering the pKa of the coordinated water.

**Keywords:** mixed metal phosphonates; metalloligand; alkaline earth metal; proton conduction





## 1. Introduction

Metal phosphonates (MPs) are a subclass of organic–inorganic hybrid materials, often combined with high water and thermal stability, showing potentially valuable properties such as ion exchange, sorption, catalysis, and proton conduction [1–5]. Incorporating mixed metal species can further expand the structural diversity of MPs and tune their properties through different metal components [6,7]. However, synthesizing mixed metal phosphonates is often challenging in terms of controlling, predicting, and crystallizing the final product. The coordination-driven self-assembly is highly influenced by numerous factors, such as temperature, pH value, solubility, coordination modes of the ligand, and other weak interactions [8]. Generally, there are three synthetic approaches to producing mixed metal phosphonates: (1) directly reacting different metal salts with ligands; (2) using ditopic or polytopic ligands that consist of two or more metal ion receptors; (3) using well-defined metal complexes as the metalloligands. The metalloligand approach is more controllable among the three approaches and has developed remarkably in the synthesis of mixed metal coordination polymers [9,10]. However, such a strategy is limited to mixed metal phosphonates due to rare phosphonate-based metalloligands compared to carboxylate-based, azole-based, and pyridine-based metalloligands [11].

In our previous work, we explored a neutral mononuclear complex $Co(notpH_3)$ based on tripodal phosphonic acid $[notpH_6 = C_9H_{18}N_3(PO_3H_2)_3]$ as the metalloligand, which can serve as a bi, tri, or tetradentate ligand to bind various metal cations such as Ag(I), Ca(II), Co(II), Ni(II), and Ln(III) [11–13]. In these $Co(notpH_x)^{3-x}$-based mixed metal phosphonates, the selected second metal ions can influence the structural dimensionality, the degree of protonation in phosphonate groups, and the amount of coordinated water. Such

variable factors agree with the design considerations (proton sources and proton transfer pathways) of intrinsic proton-conductive materials and are ideal for systematically understanding the relationship between proton conductivity and structure [14,15]. Recently, isostructural $[M^{II}_3Co^{III}_2(notp)_2(H_2O)_{12}] \cdot 2H_2O$ (M = Co or Ni) provided an example of the effect of hydrated metal ions on proton conduction, demonstrating the enhancement of proton conductivity as a result of the stronger Lewis acidity of Co(II) over Ni(II) [13]. As we know, the Lewis acid strengths of divalent alkaline earth metal ions follow the tendency: Be > Mg > Ca > Sr > Ba [16], and their unusual coordination geometries are often observed [17]. The reported $[CoCa(notpH_2)(H_2O)_2]ClO_4 \cdot nH_2O$ (**CoCa·nH₂O**) shows a moderate proton conductivity of $1.55 \times 10^{-5}$ S cm$^{-1}$ at 25 °C and 95% relative humidity (RH) [15]. It would be interesting to further study how the various alkaline earth metal nodes affect the topology and proton-conducive properties in related compounds. Herein, we use the metalloligand Co(notpH₃) to react with Mg(II), Sr(II), and Ba(II) ions to obtain three new Co-M phosphonates: $[CoMg(notpH_2)(H_2O)_2]ClO_4 \cdot nH_2O$ (**CoMg·nH₂O**), $[Co_2Sr_2(notpH_2)_2(H_2O)_5](ClO_4)_2 \cdot nH_2O$ (**CoSr·nH₂O**), and $[CoBa(notpH_2)(H_2O)_{1.5}]ClO_4$ (**CoBa**) (Figure 1). The synthesis, crystal structures, and proton-conductive properties of these compounds are reported.

**Figure 1.** Synthesis of compounds CoMg·nH₂O, CoSr·nH₂O, and CoBa.

## 2. Materials and Methods

The metalloligand Co(notpH₃)·3H₂O was synthesized using the literature procedure [12]. All other starting materials, reagents, and solvents were obtained from commercial suppliers and were used without further purification. The infrared spectra were recorded on a Bruker Tensor 27 spectrometer using KBr pellets, and the powder X-ray diffraction patterns were obtained with a Bruker D8 advance diffractometer using Cu-$K_\alpha$ radiation ($\lambda$ = 1.5406 Å). Thermogravimetric analyses (TGA) were conducted with a Mettler Toledo TGA/DSC instrument from 25 to 500 °C, with a heating rate of 5 °C min$^{-1}$ under a nitrogen atmosphere. The conductivities of the sample pellets were obtained by AC impedance measurements, which were carried out under different environmental conditions by the conventional quasi-four-probe method with a Solartron SI 1260 Impedance/Gain-Phase Analyzer and 1296 Dielectric Interface in the frequency range 1.0 MHz–0.1 Hz. The electrical contacts were prepared using the gold paste to attach the 50 μm-diameter gold wires to the 2.5 mm-diameter compressed pellets or selected single crystals. Exposure of the samples to a humid environment (40% to 95%) at different temperatures (15 to 45 °C) was performed using a GSJ-100 (Su-Ying Corp.) humidity-controlled oven.

### 2.1. Synthesis and Crystallization of [CoMg(notpH₂)(H₂O)₂]ClO₄·nH₂O (CoMg·nH₂O)

Mg(OH)₂ (0.60 mmol, 35 mg) was added to a solution of Co(notpH₃)·3H₂O (0.20 mmol, 104 mg) in water (10 mL). The suspension was stirred overnight at 100 °C and then filtered. A concentration of 1.0 mol/L HClO₄ adjusted the filtrate to pH 3.0; this was then left at room temperature for one week to afford the violet rectangular plate-like crystals of

**CoMg nH₂O** at 34% (based on Co). IR (KBr, cm$^{-1}$): 3420(br), 2922(s), 2850(m), 2403(w), 1655(m), 1487(w), 1467(w), 1420(w), 1304(w), 1286(w), 1252(w), 1215(w), 1140(s), 1120(w), 1078(s), 1053(w), 1028(s), 1007(w), 968(w), 949(m), 864(w), 802(w), 781(m), 756(w), 625(m), 608(s), 526(w), 500(w), 488(w), 438(w).

*2.2. Synthesis and Crystallization of [Co₂Sr₂(notpH₂)₂(H₂O)₅](ClO₄)₂·nH₂O (CoSr·nH₂O)*

This compound was synthesized following a similar procedure to that of **CoMg**, except that Sr(OH)₂ (0.60 mmol, 73 mg) was used, and the filtrate was adjusted to pH 2.4. The violet tetragonal crystals of **CoSr·nH₂O** were precipitated from the filtrate and collected after two days. Yield: 70% (based on Co). IR (KBr, cm$^{-1}$): 3410(br), 2995(m), 2923(m), 2854(w), 1634(m), 1493(w), 1468(w), 1420(w), 1306(w), 1284(w), 1254(w), 1194(m), 1163(m), 1144(m), 1121(s), 1107(s), 1090(s), 1026(m), 995(s), 953(m), 924(w), 843(w), 804(w), 781(w), 758(w), 609(s), 586(m), 525(w), 492(w), 436(w).

*2.3. Synthesis and Crystallization of [CoBa(notpH₂)(H₂O)₃]ClO₄ (CoBa)*

This compound was synthesized following a similar procedure to that of **CoMg**, except that Ba(OH)₂ (0.60 mmol, 103 mg) was used, and the filtrate was adjusted to pH 2.5. The violet hexagonal plate-like crystals of **CoBa** were precipitated from the filtrate and collected after one day. Yield: 77% (based on Co). IR (KBr, cm$^{-1}$): 3435(br), 2976(w), 2929(w), 2858(w), 2414(w), 1637(m), 1473(w), 1445(w), 1306(w), 1249(w), 1211(m), 1188(w), 1153(w), 1121(s), 1107(s), 1065(s), 1020(m), 1002(s), 955(w), 926(m), 854(w), 804(w), 783(m), 756(w), 607(m), 586(w), 484(w), 449(w).

*2.4. Structure Determinations*

For **CoMg·4H₂O** and **CoBa**, single crystals with dimensions of 0.30 × 0.10 × 0.05 mm$^3$ and 0.30 × 0.30 × 0.05 mm$^3$ were respectively selected and sealed in the mother solution for data collection on a Bruker SMART APEX II diffractometer using graphite-monochromated Mo-$K_\alpha$ radiation ($\lambda$ = 0.71073 Å) at room temperature (296 K). For **CoSr·2H₂O**, a single crystal with dimensions of 0.40 × 0.10 × 0.10 mm$^3$ was used for data collection on a Bruker D8 diffractometer using graphite-monochromated Mo-$K_\alpha$ radiation ($\lambda$ = 0.71073 Å) at 123 K. The numbers of collected and observed independent [$I > 2\sigma(I)$] reflections were 16662 and 5773 ($R_{int}$ = 0.075) for **CoMg·4H₂O**, 16662 and 11247 ($R_{int}$ = 0.054) for **CoSr·2H₂O**, and 10899 and 1384 ($R_{int}$ = 0.049) for **CoBa**. The data were integrated using the Siemens SAINT program [18]. Adsorption corrections were applied. The structures were solved by direct methods and refined on $F^2$ by full-matrix least-squares using SHELXTL [19,20]. Anisotropic temperature factors were used to refine all atoms, excluding hydrogen. All hydrogen atoms bound to carbon were refined isotropically in the riding mode; hydrogen atoms of water molecules were detected in the experimental electron density and then refined isotropically with reasonable restriction of O-H bond distances and H-O-H angles. The crystallographic data are given in Table S1, and selected bond lengths and angles are in Tables S2–S4.

## 3. Results

*3.1. Structural Describes*

A single crystal of **CoMg·4H₂O** was sealed in the mother solutions to keep the saturated lattice water content, and these were used for the X-ray single-crystal structural determination at room temperature. Structural analysis reveals that **CoMg·4H₂O** is isostructural to **CoCa·4H₂O** [15]. It crystallizes in monoclinic system space groups $P2_1/n$ (Table S1) and contains one [Co(notpH₂)]$^-$ ligand, one Mg$^{2+}$, one ClO₄$^-$, and two coordinated and four lattice water molecules in the asymmetric unit (Figure 2a). The [Co(notpH₂)]$^-$ links four Mg atoms via the phosphonate oxygen atoms O2, O5, O8, and O9 as a tetradentate metalloligand. The two phosphonate oxygen atoms (O3 and O6) of the [Co(notpH₂)]$^-$ unit are protonated. Each Mg atom is six-coordinated, with four sites provided by four phosphonate oxygens and two water molecules. The average Mg-O bond length is 2.108(3) Å

[2.031(3)–2.253(4) Å], which is shorter than the average Ca-O bond length of 2.353(3) Å [2.273(3)–2.477(3) Å]. The {CoN₃O₃} and {MgO₆} octahedra are each corner-shared with the {PO₃C} tetrahedra, forming a positively charged two-dimensional waved layer containing 8- and 16-member rings (Figure 2b). The positively charged layers are charge-balanced by $ClO_4^-$ anions. The interlayer spaces are filled with the $ClO_4^-$ anions and the lattice water molecules (Figure 2c). **CoMg·4H₂O** has the same layer topology and hydrogen bond networks as **CoCa·4H₂O**, including the protonated sites of the phosphonate oxygen atoms.

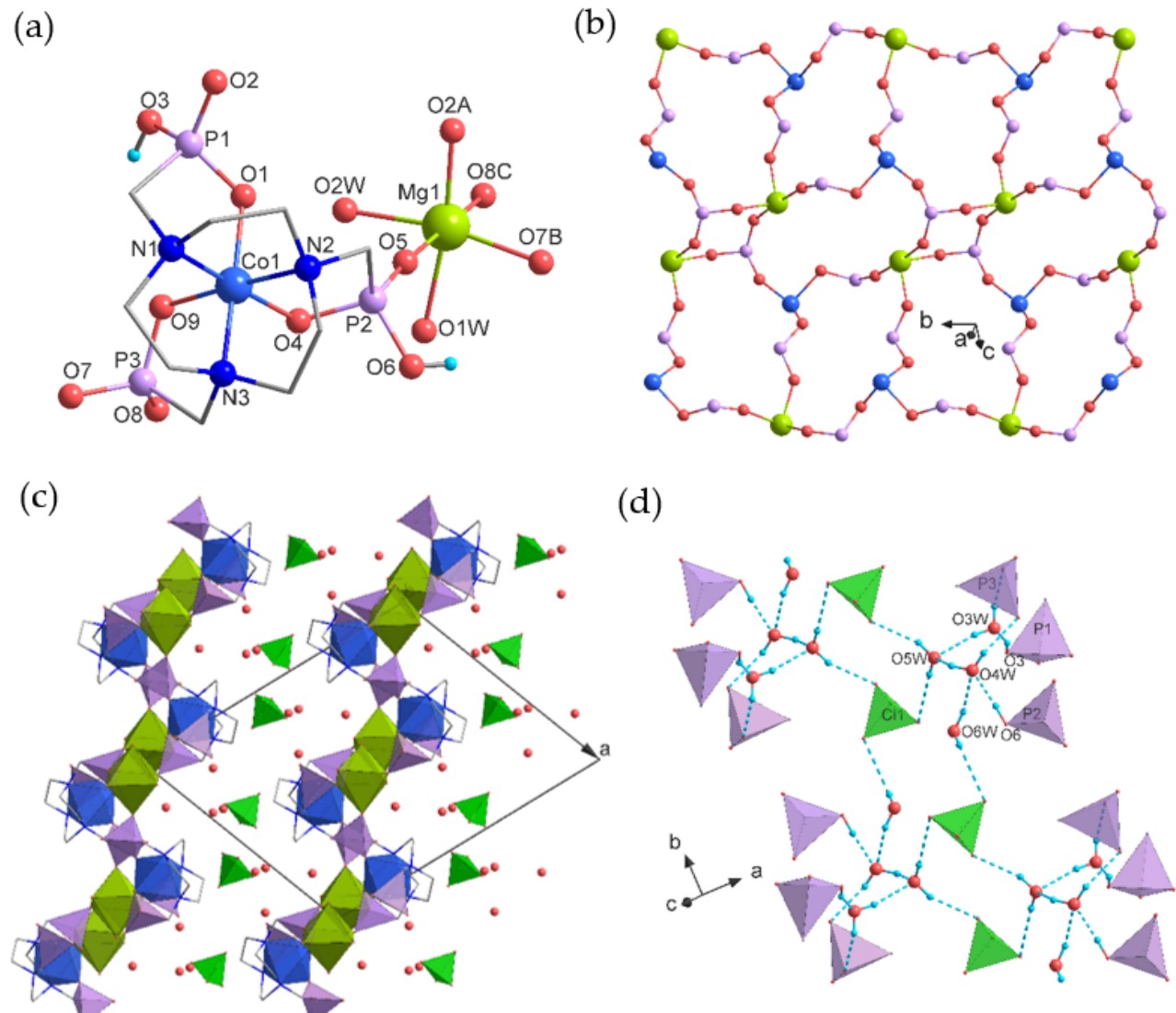

**Figure 2.** The crystal structure of **CoMg·4H₂O**; the asymmetric unit (**a**), one inorganic layer (**b**), the three-dimensional packing diagram (**c**), and hydrogen chains composed of lattice water molecules, phosphonate groups, and perchlorate anions (**d**). All hydrogen atoms are omitted for clarity, except in the protonated phosphonate groups or the water molecules.

Unlike **CoMg·4H₂O** and **CoCa·4H₂O**, when the Co(notpH₃) ligand is assembled with larger $Sr^{2+}$ and $Ba^{2+}$ ions, the resulting products are crystallized with distinct coordination geometries and layered structures. The compound **CoSr·2H₂O** crystallized in the triclinic system space group *P*-1 (Table S1). It contains two [Co(notpH₂)]⁻, two $Sr^{2+}$, two $ClO_4^-$, and five coordinated and three lattice water molecules (occupancy: O6W 1.0, O7W 0.5, and O8W 0.5) in the asymmetric unit (Figure 3a). Each [Co(notpH₂)]⁻ unit has two protonated phosphonate oxygen atoms (O3 and O9 in Co1 unit; O12 and O19 in Co2 unit). The

[Co1(notpH$_2$)]$^-$ behaves as a tetradentate metalloligand and connects four Sr atoms by four phosphonate oxygen atoms, O1, O2, O4, and O8. In contrast, the [Co2(notpH$_2$)]$^-$ behaves as a tridentate metalloligand and connects three Sr atoms by three phosphonate oxygen atoms, O10, O11, and O14. The Sr1 atom is seven-coordinated with two water molecules and five phosphonate oxygen atoms (O2, O4, O4A, O8B, and O10) from the three [Co1(notpH$_2$)]$^-$ and single [Co2(notpH$_2$)]$^-$ ligands. The Sr2 atom also adopts a seven-coordinated geometry but is surrounded by three water molecules and four phosphonate oxygen atoms (O1, O11, O14, and O14C) from one [Co1(notpH$_2$)]$^-$ and two [Co2(notpH$_2$)]$^-$ ligands. The average Sr-O bond length is 2.574(5) Å [2.453(5)–2.659(5) Å] for Sr1 and 2.591(5) Å [2.472(5)–2.693(5) Å] for Sr2. Each pair of seven-coordinated Sr1 or Sr2 atoms are bridged by two oxygen atoms (O4 and O4A, or O14 and O14C) to form a Sr1$_2$O$_2$ or Sr2$_2$O$_2$ dimeric unit, and such dimeric units are further alternately connected by a pair of O-P-O to form an inorganic chain. The {Sr1O$_7$} units within the adjacent inorganic chains are interconnected via a pair of O-Co1-O-P-O, resulting in a positively charged two-dimensional layer (Figure 3b). The ClO$_4$$^-$ anions and the lattice water molecules fill the interlayer spaces (Figure 3c).

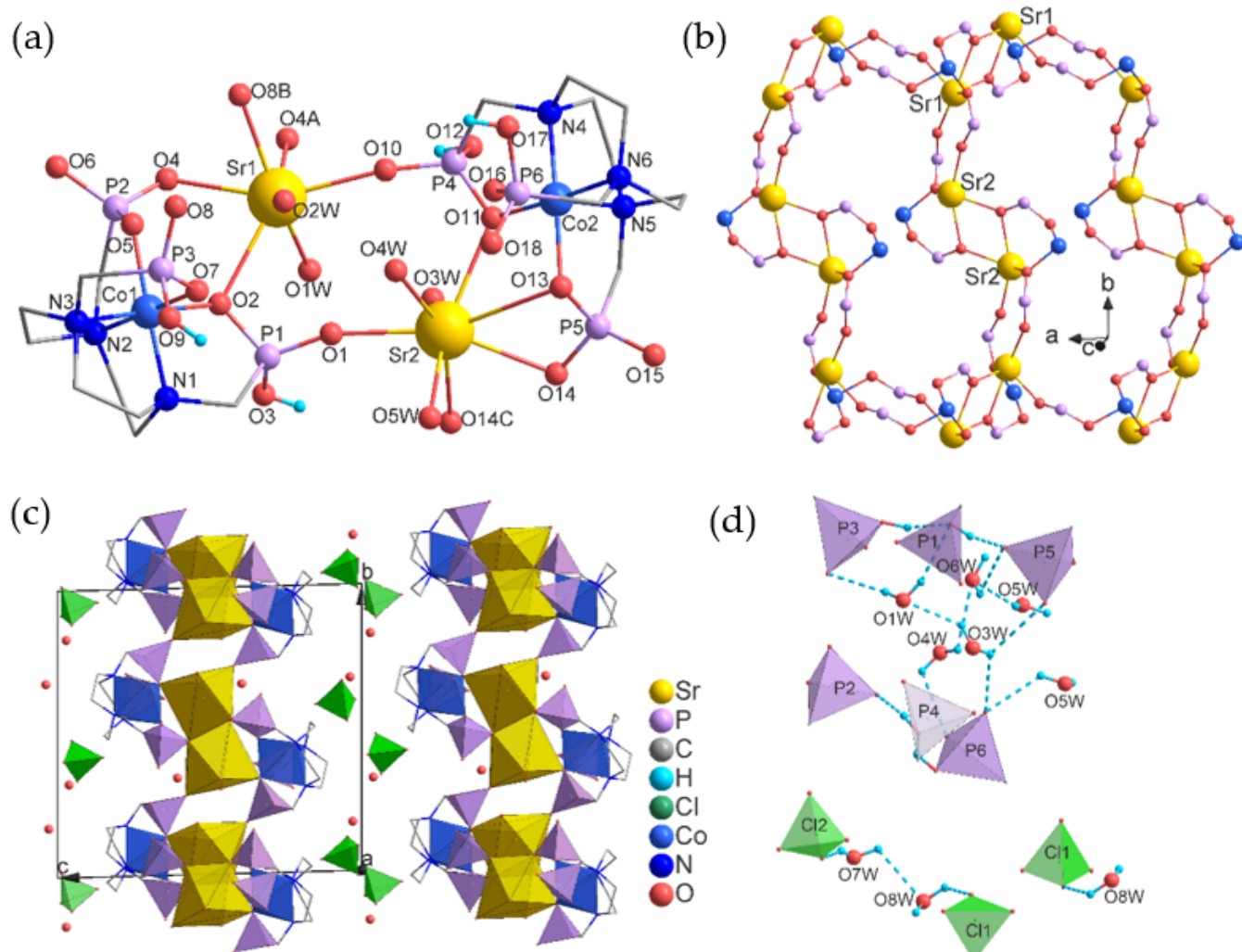

**Figure 3.** The crystal structure of **CoSr·2H$_2$O**; the asymmetric unit (**a**), one inorganic layer (**b**), the three-dimensional packing diagram (**c**), and hydrogen bonds among the water molecules, phosphonate groups, and perchlorate anions (**d**). All hydrogen atoms are omitted for clarity, except in the protonated phosphonate groups or water molecules.

The crystal's space group of **CoBa** is *R*-3*c* with the hexagonal unit cell of *a* = 8.5991(15) and *c* = 97.615(18) Å. The asymmetric unit contains 1/3 [Co(notpH$_2$)]$^-$, 1/3 Ba$^{2+}$, 1/3 ClO$_4$$^-$,

and a half-coordinated water molecule (Figure 4a). Each Co, Ba, and Cl atom is located at a three-fold rotation axis parallel to the *c*-axis. Furthermore, each coordinated water molecule is located at a two-fold rotation axis parallel to the *a*-axis. In the $[Co(notpH_2)]^-$ unit, two of the three equivalent phosphonate groups should be monoprotonated according to the charge balance. Therefore, the occupancy of the H atom in the asymmetric -$PO_3H$ group is set as 2/3. The $[Co(notpH_2)]^-$ behaves as a hexadentate metalloligand, which chelates one Ba atom with O1, O1A, and O1B and connects another three Ba atoms via O2, O2A, and O2B. The Ba atom is coordinated to six phosphonate oxygen atoms (O1, O1A, O1B, O2C, O2D, and O2E) from four $[Co(notpH_2)]^-$ ligands and three water molecules (O1W, O1WA, and O2WB), showing a distorted tricapped trigonal prismatic geometry. The average Ba-O bond length is 2.853(10) Å [2.724(13)–2.948(8) Å]. The $[Ba(H_2O)_{1.5}]_n$ layer with a hexagonal grid extends in the *ab* plane (Figure 4b), where both sides are covered by the $[Co(notpH_2)]^-$ units. The positive coordination layers $[CoBa(notpH_2)(H_2O)_{1.5}]$ exhibit an ABCABC type of three-dimensional packing and are charge balanced by the lattice $ClO_4^-$ anions (Figure 4c).

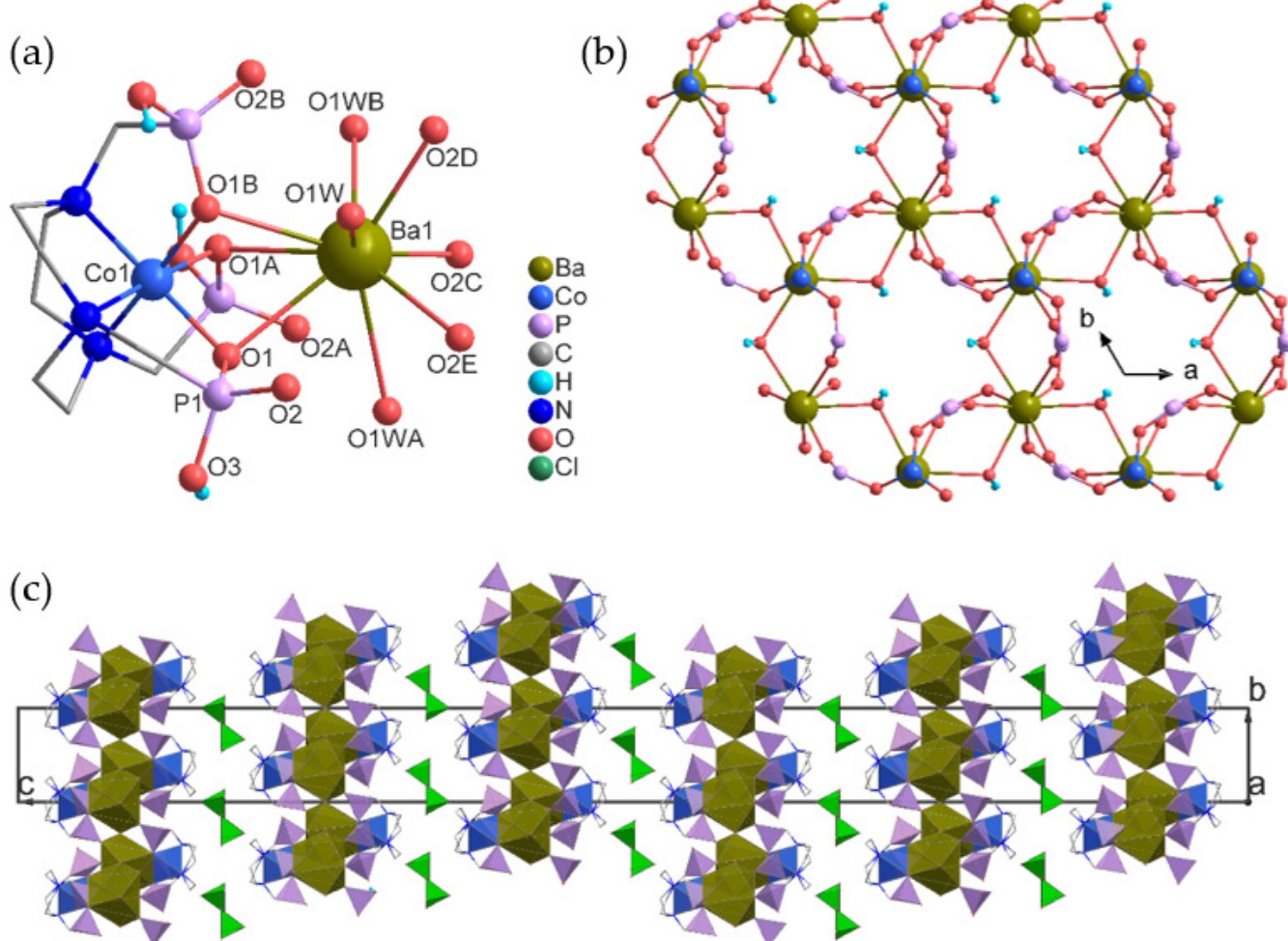

**Figure 4.** The crystal structure of **CoBa**; the asymmetric unit (**a**), one hexagonal inorganic layer (**b**), and the ABCABC type of the three-dimensional packing diagram (**c**). All hydrogen atoms are omitted for clarity, except in the protonated phosphonate groups or water molecules.

## 3.2. Thermal Stability

The thermogravimetric analyses were performed on **CoMg·nH₂O**, **CoSr·nH₂O**, and **CoBa** to compare their thermal stabilities. As shown in Figure 5, the **CoMg·nH₂O** stored in the air (ca. 50% RH) shows a two-step decomposition process from 25 to 500 °C. Dehydration occurs below 130 °C, and the weight loss of 11.2% agrees with releasing

two lattice water and two coordinated water molecules (calc. 10.9% for **CoMg·2H₂O**). It indicates that the collected product of **CoMg·nH₂O** lost two lattice water molecules per formula unit in the air and formed the dihydrate phase, **CoMg·2H₂O**, which is confirmed by the Pawley fitting of the PXRD pattern (Figure S1). The fitted unit cell parameters of **CoMg·2H₂O** ($P2_1/n$, $a$ = 13.14 Å, $b$ = 9.93 Å, $c$ = 18.58 Å, $\beta$ = 109.2°, $V$ = 2289.3 Å³) are identical to those of the reported compound, **CoCa·2H₂O**. Following a weight-loss plateau between 130 and 270 °C, the rapid weight loss is attributed to the pyrolysis of the organic moieties. The water loss of **CoSr·nH₂O** occurs below 120 °C, and the weight loss of 8.4% agrees with the release of two lattice water and five coordinated water molecules (calc. 8.8%). A broad weight-loss plateau starts from 120 °C to 280 °C, followed by rapid weight loss due to the pyrolysis of organic moieties. **CoBa** shows high thermal stability without mass change up to 270 °C. Above 270 °C, the removal of coordinated water and the pyrolysis of organic moieties happens simultaneously.

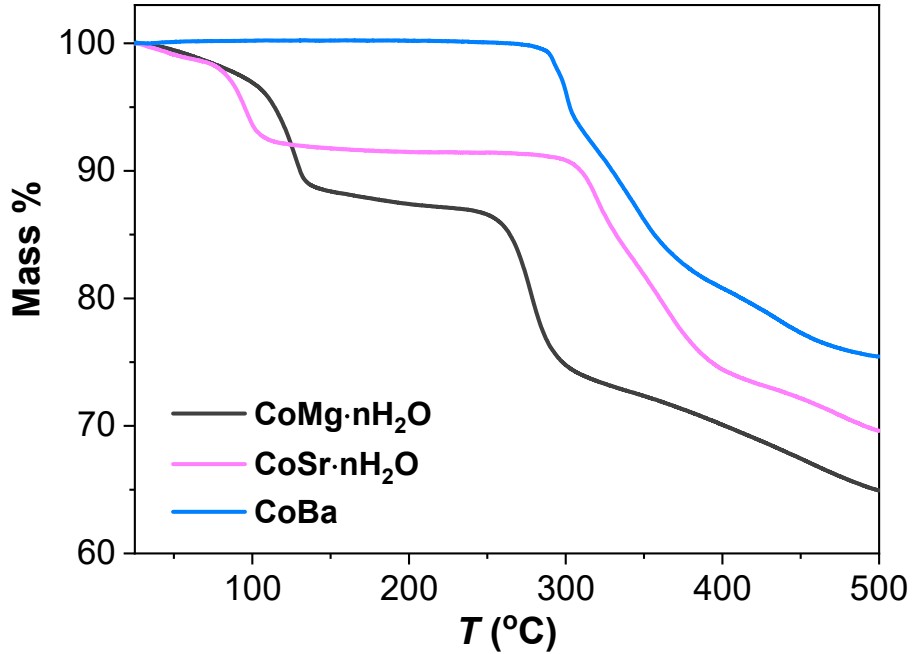

**Figure 5.** The thermogravimetric curves of **CoMg·nH₂O**, **CoSr·nH₂O**, and **CoBa**.

### 3.3. Proton Conduction

The proton conductivities of **CoMg·nH₂O**, **CoSr·nH₂O**, and **CoBa** were evaluated by impedance measurements (Figures S5–S13) using pressed pellets (1.0 Gpa) with thicknesses of 0.76, 0.75, and 0.69 mm, respectively. As shown in Figure 6a, all three samples and the reported compound, **CoCa·nH₂O**, exhibit humidity-dependent proton-conductivities under 25 °C. At 40% RH, the proton conductivities of **CoMg·nH₂O**, **CoCa·nH₂O**, **CoSr·nH₂O**, and **CoBa** are $1.83 \times 10^{-7}$, $1.08 \times 10^{-9}$, $1.14 \times 10^{-8}$, and $4.28 \times 10^{-8}$ S cm⁻¹, respectively. All samples' conductivities increased with relative humidity, reaching maximum values at 95% RH. Furthermore, the conductivity at 95 % RH and 25 °C followed the sequence: **CoMg·nH₂O** ($4.36 \times 10^{-4}$ S cm⁻¹) > **CoCa·nH₂O** ($1.55 \times 10^{-5}$ S cm⁻¹) > **CoBa** ($1.31 \times 10^{-5}$ S cm⁻¹) > **CoSr·nH₂O** ($3.02 \times 10^{-6}$ S cm⁻¹).

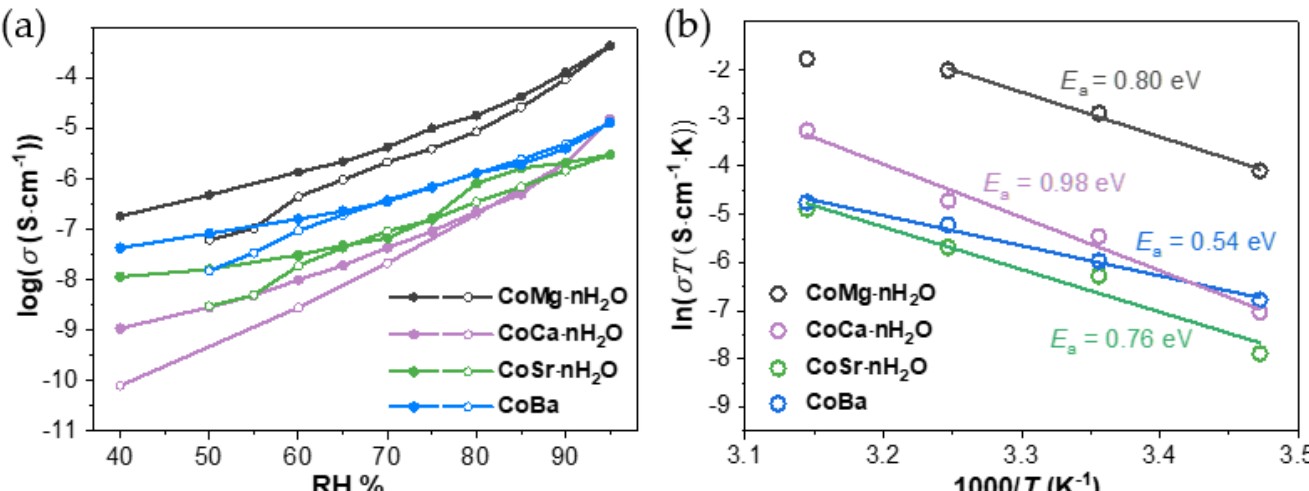

**Figure 6.** Proton-conductive properties of **CoMg·nH₂O**, **CoSr·nH₂O**, and **CoBa**; (**a**) plots of log($\sigma$) vs. RH at 25 °C (increasing RH: filled circle; decreasing RH: open circle). (**b**) Plots of ln($\sigma T$) vs. 1000/$T$ at 95% RH. The data of **CoCa·nH₂O** are taken from Bao et al. [15].

The temperature dependence of the conductivities was measured at 95% RH from 15 to 45 °C at 10 °C intervals. Figure 6b shows the ln($\sigma T$) plots vs. 1000/$T$ for all samples. The activation energies, $E_a$, are estimated to be 0.80 eV for **CoMg·nH₂O**, 0.76 eV for **CoSr·nH₂O**, and 0.54 eV for **CoBa**. **CoMg·4H₂O** has a continuous hydrogen-bonding network involving $ClO_4^-$ anions, -$PO_3H$ groups, and water molecules. Therefore, the large proton conduction activation energy of **CoMg·4H₂O** can arise from the rotation and movement of the $ClO_4^-$ anions, as is the case for **CoCa·4H₂O**. In **CoSr·nH₂O** and **CoBa**, the hydrogen bonds are isolated, and the proton might migrate through the medium via the vehicle-type mechanism, corresponding to the large activation energies.

## 4. Discussion

### 4.1. The Effects of Varying the Alkaline Earth Metal Nodes on the Structures

It could be interesting to compare the different coordination modes of tripodal metalloligand Co(notpH$_x$)$^{x-3}$ with Mg(II), Ca(II), Sr(II), and Ba(II), respectively (Figure 7). Steric factors commonly govern the coordination geometry of the alkaline earth metal cations. In **CoSr·2H₂O**, 7-coordinate {SrO₇} is the distorted capped triangular prism. The smaller Mg(II) and Ca(II) ions are 6-coordinated to form the distorted octahedron in **CoMg·4H₂O** and **CoCa·4H₂O**. In contrast to Sr(II), the larger Ba(II) ion is 9-coordinated to form the distorted tricapped triangular prism {BaO₉} in **CoBa**. Moreover, water molecules occupy three vertices on the prism's three triangular faces. The coordination number increases with the radius of the alkaline earth metal ions; consequently, the tripodal metalloligand Co(notpH$_x$)$^{x-3}$ adopts variable bonding modes. In **CoMg·4H₂O** and **CoCa·4H₂O**, the Co(notpH$_2$)$^-$ offers four oxygen atoms to bind four Mg(II) or Ca(II) ions, and the other two oxygen atoms are protonated. In **CoSr·2H₂O**, two coordination modes for Co(notpH$_2$)$^-$ are observed, and two μ-O atoms bridge the two Sr(II) ions and adjacent Co(III) and Sr(II) ions, respectively. Interestingly, Co(notpH$_2$)$^-$ chelates the Ba(II) ion in **CoBa** using three oxygen atoms, which are coordinated with the Co(III) ion. Furthermore, each equivalent phosphonate group of Co(notpH$_3$) is monoprotonated (occupancy of H is 2/3), and the other three oxygen atoms bind to three Ba(II) ions. Such a coordination mode differs from the reported Co(notpH$_x$)$^{x-3}$-based mixed metal phosphonates [11–13].

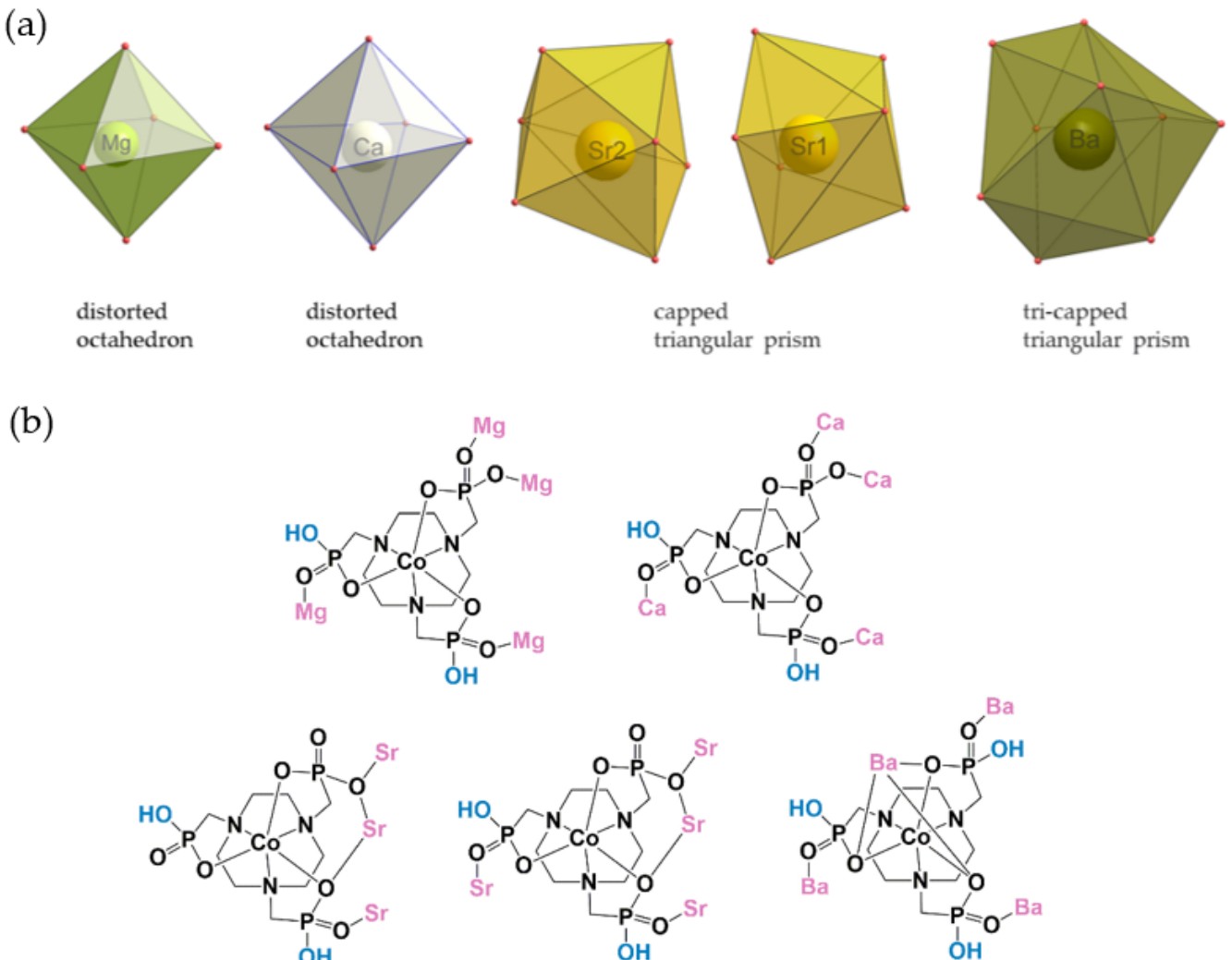

**Figure 7.** Diversities of coordination geometry (**a**) at the Mg(II), Ca(II), Sr(II), and Ba(II) centers, and coordination modes (**b**) in the tripodal metalloligand Co(notpH$_2$)$^-$.

*4.2. The Effects of Varying the Alkaline Earth Metal Nodes on Proton Conduction*

Sufficient acidic proton concentration and continuous hydrogen-bonding networks play critical roles in efficient proton conduction. Coordination water, the acidic moieties of the frameworks, and acidic guest molecules can act as proton sources. For **CoMg·nH$_2$O**, **CoCa·nH$_2$O**, **CoSr·nH$_2$O**, and **CoBa**, the equal number of protonated phosphonate oxygen atoms is 2 per ligand, and the numbers of coordinated water are 2, 2, 2.5, and 1,5, respectively. However, the lack of continuous hydrogen bonds in **CoSr·nH$_2$O** and **CoBa** leads to poorer proton conduction (Figure 3d). It is worth noting that **CoBa** contains no lattice water but exhibits humidity-dependent proton conduction, which could be attributed to the effect of the grain boundary using a pellet for measurement [21]. In isostructural **CoMg·nH$_2$O** and **CoCa·nH$_2$O**, the proton sources and proton pathways are identical (Figure 8), but **CoMg·nH$_2$O** exhibits a 28-fold enhanced value for proton conductivity compared with **CoCa·nH$_2$O** at 95% RH and 25 °C. The only difference between the structures of both of the compounds is the different dihydrated metal centers: Mg(II) and Ca(II). The p$K_a$ values of the aqueous metal ions Mg(H$_2$O)$_6$$^{2+}$ and Ca(H$_2$O)$_7$$^{2+}$ are 11.2 and 12.7 [22], respectively. This indicates that the water binding to the Mg(II) ion can provide more acidic protons, agreeing with the higher proton conductivity of **CoMg·nH$_2$O**.

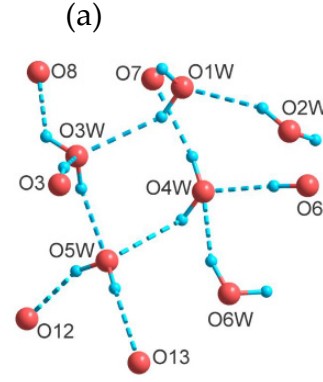

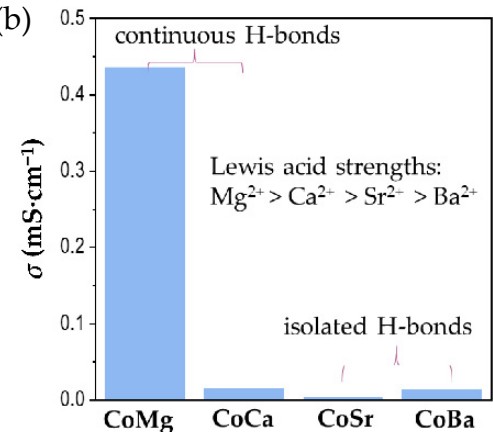

| D...A distance (Å) | CoMg·4H₂O | CoCa·4H₂O |
|---|---|---|
| O1W...O2W | 3.000(6) | 2.865(4) |
| O1W...O3W | 2.867(6) | 2.859(5) |
| O3W...O8 | 2.849(5) | 2.805(5) |
| O3W...O3 | 2.600(5) | 2.585(4) |
| O3W...O5W | 2.779(7) | 2.761(6) |
| O4W...O5W | 2.798(7) | 2.914(6) |
| O4W...O6 | 2.523(6) | 2.563(4) |
| O4W...O6W | 3.147(14) | 3.040(12) |
| O5W...O12 | 2.868(6) | 2.824(9) |
| O5W...O13 | 3.240(16) | 3.124(10) |

**Figure 8.** (**a**) Identical hydrogen-bonding networks of **CoMg·4H₂O** and **CoCa·4H₂O**. (**b**) Comparison of proton conductivities of all compounds and possible contributing factors.

## 5. Conclusions

We utilized the metalloligand Co(notpH₃) to obtain a series of layered Co(III)-M(II) phosphonates (M = Mg, Sr, and Ba). The layer topology of **CoMg·4H₂O** is identical to that of **CoMg·4H₂O**, whereas **CoSr·2H₂O** and **CoBa** exhibit diverse layer topologies. Especially in **CoBa**, where the Co(notpH₂)⁻ chelates the Ba(II) ion using three oxygen atoms, which are also coordinated with the Co(III) ion, specializing in the other reported Co(notpHₓ)ˣ⁻³-based mixed metal phosphonates. The proton conductivities of all compounds were evaluated using compacted pellets and followed the sequence (at 25 °C and 95% RH): **CoMg·nH₂O** (4.36 × 10⁻⁴ S cm⁻¹) > **CoCa·nH₂O** (1.55 × 10⁻⁵ S cm⁻¹) > **CoBa** (1.31 × 10⁻⁵ S cm⁻¹) > **CoSr·nH₂O** (3.02 × 10⁻⁶ S cm⁻¹). There are no continuous hydrogen-bonding networks in **CoSr·2H₂O** and **CoBa**, resulting in poor proton conduction. **CoMg·nH₂O** and **CoCa·nH₂O** are isostructural except for the Mg(II) and Ca(II) centers and have identical hydrogen-bonding networks as a proton transfer pathway. However, the Mg(II) center shows greater Lewis acid strength and makes the coordinated water more acidic, leading to a 28-fold enhanced proton conductivity in **CoMg·nH₂O** at 25 °C and 95% RH.

**Supplementary Materials:** The following supporting information can be downloaded at: https://www.mdpi.com/article/10.3390/cryst12111648/s1, Table S1: Crystallographic data for **CoMg·4H₂O**, **CoSr·2H₂O**, and **CoBa**; Table S2: Selected bond lengths (Å) and angles (°) for **CoMg·4H₂O**; Table S3: Selected bond lengths (Å) and angles (°) for **CoSr·2H₂O**; Table S4: Selected bond lengths (Å) and angles (°) for **CoBa**; Figure S1: (a) Comparison of the observed and simulated powder X-ray diffraction patterns of **CoMg·nH₂O**. (b) The observed pattern is fitted by the Pawley method using Topas 5.0 program; Figure S2: (a) Comparison of the observed and simulated powder X-ray diffraction patterns of **CoSr·nH₂O**. (b) The observed pattern is fitted by the Pawley method using Topas 5.0 program; Figure S3: Comparison of the observed and simulated powder X-ray diffraction patterns of **CoBa**; Figure S4: IR spectra of **CoMg·nH₂O**, **CoSr·nH₂O**, and **CoBa**; Figure S5: Nyquist plots for the pellet of **CoMg·nH₂O** at 25 °C and various RH; Figure S6: Nyquist plots for the pellet of **CoMg·nH₂O** at 25 °C and various RH; Figure S7: Nyquist plots for the pellet of **CoMg·nH₂O** at 95% RH and various temperatures; Figure S8: Nyquist plots for the pellet of **CoSr·nH₂O** at 25 °C and various RH; Figure S9: Nyquist plots for the pellet of **CoSr·nH₂O** at 25 °C and various RH; Figure S10: Nyquist plots for the pellet of **CoSr·nH₂O** at 95% RH and various temperatures; Figure S11: Nyquist plots for the pellet of **CoBa** at 25 °C and various RH; Figure S12: Nyquist plots for the pellet of **CoBa** at 25 °C and various RH; Figure S13: Nyquist plots for the pellet of **CoBa** at 95% RH and various temperatures.

**Author Contributions:** Conceptualization, S.-S.B.; methodology, S.-S.B.; software, S.-S.B.; validation, N.-Z.L., Y.-X.W. and S.-S.B.; formal analysis, N.-Z.L., Y.-X.W. and. Y.S.; investigation, N.-Z.L., Y.-X.W., Y.S. and S.-S.B.; resources, S.-S.B.; data curation, S.-S.B.; writing—original draft preparation, S.-S.B.; writing—review and editing, N.-Z.L., Y.-X.W., Y.S. and S.-S.B.; visualization, S.-S.B.; supervision, S.-S.B.; project administration, S.-S.B.; funding acquisition, S.-S.B. All authors have read and agreed to the published version of the manuscript.

**Funding:** This work was funded by the National Natural Science Foundation of China (21671098, 21731003) and the Fundamental Research Funds for the Central Universities (14380151, 14380206).

**Data Availability Statement:** CCDC 2214708–2214710 contains the supplementary crystallographic data for this paper. These data can be obtained free of charge from The Cambridge Crystallographic Data Centre via www.ccdc.cam.ac.uk/data_request/cif, accessed on 12 November 2022.

**Conflicts of Interest:** The authors declare no conflict of interest.

**Abbreviations**

The following abbreviations are used in this manuscript:

| | |
|---|---|
| MP | metal phosphonate |
| notpH$_6$ | 1,4,7-Triazacyclononane-1,4,7-triyl-tris(methylene-phosphonic acid |
| RH | relative humidity |

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
