# Peer review of "Mixed Metal Phosphonates: Structure and Proton Conduction Manipulation through Various Alkaline Earth Metal Ions"

_crystals, doi:10.3390/cryst12111648_

Round 1
Reviewer 1 Report
The paper of Song-Song Bao and co-authors present the synthesis, physicochemical characterisation, and study of proton conduction of three new bimetallic phosphonates. These materials are very interesting, because play a central role in energy technologies (energy storage, renewable energy technologies, etc.). In general, the article is well written, but some details are not clear.
I would like to address only some minor issues:
1) Between data (formula and molar weight) for CoSr·2H2O mentioned in the section „Structure describes“ and Table S1 (Supplementary Materials) are some discrepancy. Please correct this discrepancy.
2) The thermal stability of studied compounds were performed in range 25 to 500 °C. Why was not performed the thermal analysis in wider range?
3) The experimental weight loss for CoMg·nH2O occurs below 130°C was 11.2% (section: Thermal stability, line 209), and it was interpreted as dehydratation of sample, releasing of two lattice water and two coordinated molecules of water (calculated weight loss 10.9% - line 210). However, the weight loss calculatede from data in Table S1 (molar weight for CoMg·4H2O) is 10.3%. Please correct this discrepancy.
4) In the part „Thermal analysis“ is a statement „It indicates that the collected product of CoMg·nH2O lost two lattice water molecules per formula unit in the air and formed the dihydrate phase CoMg·2H2O, which is confirmed by the XPRD pattern (Figure S1)...“. This statement was based on comparison of cell parameters of isostructural CoCa·2H2O or another facts? Please clarify it in the main text.
In my opinion, this work sounds reasonable and I could recommend the publication of this work in Crystals.
Author Response
The paper of Song-Song Bao and co-authors present the synthesis, physicochemical characterisation, and study of proton conduction of three new bimetallic phosphonates. These materials are very interesting, because play a central role in energy technologies (energy storage, renewable energy technologies, etc.). In general, the article is well written, but some details are not clear.
Response: We appreciate the reviewer for the positive and encouraging comments.
I would like to address only some minor issues:
1) Between data (formula and molar weight) for CoSr·2H2O mentioned in the section "Structure describes" and Table S1 (Supplementary Materials) are some discrepancy. Please correct this discrepancy.
Response: Thanks for this question. For CoSr·nH2O, single crystal XRD analysis revealed that the asymmetric unit contains three lattice water molecules, but their total occupancy is 2.0 (O6W 1.0, O7W 0.5, and O8W 0.5). Therefore, the single-crystal sample is named [Co2Sr2(notpH2)2(H2O)5](ClO4)2·2H2O (CoSr·2H2O) with a formula of C18H54N6O33P6Cl2Co2Sr2 (Table S1). And the collected polycrystalline sample is named [Co2Sr2(notpH2)2(H2O)5](ClO4)2·nH2O (CoSr·nH2O) due to the humidity-dependent amount of lattice water.
2) The thermal stability of studied compounds were performed in range 25 to 500 °C. Why was not performed the thermal analysis in wider range?
Response: Thanks for this question. For CoMg·nH2O, CoSr·nH2O, and CoBa, the pyrolysis of organic moieties and the framework collapse happen above 280 °C. In addition, these samples contain perchlorate anions, which may cause explosions under high temperatures. Therefore, we did not perform the thermal analysis for these samples above 500 °C.
3) The experimental weight loss for CoMg·nH2O occurs below 130°C was 11.2% (section: Thermal stability, line 209), and it was interpreted as dehydratation of sample, releasing of two lattice water and two coordinated molecules of water (calculated weight loss 10.9% - line 210). However, the weight loss calculated from data in Table S1 (molar weight for CoMg·4H2O) is 10.3%. Please correct this discrepancy.
Response: Thanks for this question. The thermal analysis indicates that the collected product of CoMg·nH2O in the air is the dihydrate phase CoMg·2H2O, which is confirmed by the PXRD pattern (Figure S1). For clarity, we have revised the sentence: "Dehydration occurs below 130 °C, and the weight loss of 11.2% agrees with releasing two lattice water and two coordinated water molecules (calc. 10.9% for CoMg·2H2O)".
4) In the part "Thermal analysis "is a statement "It indicates that the collected product of CoMg·nH2O lost two lattice water molecules per formula unit in the air and formed the dihydrate phase CoMg·2H2O, which is confirmed by the XPRD pattern (Figure S1)... ". This statement was based on comparison of cell parameters of isostructural CoCa·2H2O or another facts? Please clarify it in the main text.
Response: Thanks for this question. We have revised the main text. "It indicates that the collected product of CoMg·nH2O lost two lattice water molecules per formula unit in the air and formed the dihydrate phase CoMg·2H2O, which is confirmed by the Pawley fitting of the PXRD pattern (Figure S1). The fitted unit cell parameters of CoMg·2H2O (P21/n, a = 13.14 Å, b = 9.93 Å, c = 18.58 Å, β = 109.2°, V = 2289.3 Å3 ) are identical to those of the reported compound CoCa·2H2O.”
Reviewer 2 Report
Manuscript describes interesting new compounds and their conduction properties. Three compounds have been prepared and analyzed according to the known procedure thus making the compounds the novelty of the manuscript. The obtained results could be interesting to the readers of the journal Crystals. However, some issues should be addressed before publication of the manuscript.
Description of the structures should not contain claims like "almost the same as". What does it mean? The same to what and/or how similar? Please do specify what is targeted by these claims.
Have you tried measuring the conductance along the appropriate crystallographic direction for the compound CoMg (as for some other compounds in previous studies)? This could give valuable information regarding its properties.
Do you have any explanation of the low yield for the CoMg compared to the other two compounds? Does any other form of the CoMg form? How have you come to the different pH needed to obtain the described compounds (3.0, 2.4 or 2.5)? Please describe in more details what happens if this value is changed. I think more synthetic information could be valuable for further scientific considerations regarding these or similar systems.
After these minor corrections, this manuscript should be ready for publication in the Crystals.
Author Response
Manuscript describes interesting new compounds and their conduction properties. Three compounds have been prepared and analyzed according to the known procedure thus making the compounds the novelty of the manuscript. The obtained results could be interesting to the readers of the journal Crystals. However, some issues should be addressed before publication of the manuscript.
Response: We appreciate the reviewer for the positive and encouraging comments.
Description of the structures should not contain claims like "almost the same as". What does it mean? The same to what and/or how similar? Please do specify what is targeted by these claims.
Response: Thanks for this suggestion. We have corrected those claims like "almost the same as" and highlighted revisions in yellow in the manuscript.
The revised sentences in the manuscript are "Structural analysis reveals that CoMg·4H2O is isostructural to CoCa·4H2O.", "CoMg·4H2O has the same layer topology and hydrogen bond networks as CoCa·4H2O, including the protonated sites of the phosphonate oxygen atoms.", “CoMg·4H2O has a continuous hydrogen-bonding network involving ClO4- anions, -PO3H groups, and water molecules.”
Have you tried measuring the conductance along the appropriate crystallographic direction for the compound CoMg (as for some other compounds in previous studies)? This could give valuable information regarding its properties.
Response: Thanks for this question. We have tried to measure the conductance using the single crystal of the compound CoMg·nH2O. However, the crystals cracked in a short time when they were picked out of the solution. We can not study the anisotropic proton conduction for CoMg·nH2O.
Do you have any explanation of the low yield for the CoMg compared to the other two compounds? Does any other form of the CoMg form? How have you come to the different pH needed to obtain the described compounds (3.0, 2.4 or 2.5)? Please describe in more details what happens if this value is changed. I think more synthetic information could be valuable for further scientific considerations regarding these or similar systems.
Response: Thanks for this question. Compounds CoMg, CoSr, and CoBa are precipitated from the aqueous solution by evaporation. CoMg is most easily resolved in water, probably due to the weak Mg-O coordination bond. Therefore, the yield for CoMg is the lowest among the three compounds.
We used the hydroxides of Mg, Sr, and Ba as sources of alkaline earth metals to react with the acidic metalloligand Co(notpH3). After reactions, the undissolved hydroxides of Mg, Sr, and Ba were removed by filtration. HClO4 was added to the filtrate to introduce perchlorate anions, and then the initial pH was measured. The pH value of the solution changed during evaporation. Fast precipitation was expected to obtain the pure phase by keeping the degree of ligand protonation. When the initial pH was higher than 3.5, no products were precipitated for several days. When the initial pH was lower than 2.0, the metalloligand Co(notpH3) was precipitated. 3.0, 2.4, and 2.5 are optimal initial pH for obtaining high-yield pure CoMg, CoSr, and CoBa, respectively.